Protection of cultured brain endothelial cells from cytokine-induced damage by α-melanocyte stimulating hormone

Harazin András 1 2
Bocsik Alexandra 1
Barna Lilla 1 2
Kincses András 1
Váradi Judit 3
Fenyvesi Ferenc 3
Tubak Vilmos 4
http://orcid.org/0000-0001-6084-6524 Deli Maria A. 1 deli.maria@brc.mta.hu
Vecsernyés Miklós 3 vecsernyes.miklos@pharm.unideb.hu
1 Institute of Biophysics, Biological Research Centre, Hungarian Academy of Sciences , Szeged , Hungary
2 Doctoral School in Biology, Faculty of Science and Informatics, University of Szeged , Szeged , Hungary
3 Department of Pharmaceutical Technology, Faculty of Pharmacy, University of Debrecen , Debrecen , Hungary
4 Creative Laboratory Ltd. , Szeged , Hungary
González-Burgos Elena
Electronic publication date: 2018 May 15
Publication date: 2018
Volume: 6
Electronic Location ID: e4774
Received 2018 Feb 21; Accepted 2018 Apr 23
Copyright: © 2018 Harazin et al.
Copyright year: 2018
Copyright holder: Harazin et al.
License: This is an open access article distributed under the terms of the Creative Commons Attribution License, which permits unrestricted use, distribution, reproduction and adaptation in any medium and for any purpose provided that it is properly attributed. For attribution, the original author(s), title, publication source (PeerJ) and either DOI or URL of the article must be cited.
License URL: https://creativecommons.org/licenses/by/4.0/

Keywords: Brain endothelial cells, Permeability, α-Melanocyte-stimulating hormone, Cytokines, Tight junction, Blood–brain barrier, Melanocortin-1 receptor, Reactive oxygen species

Funding: National Research, Development and Innovation Office EFOP-3.6.1-16-2016-00022, GINOP-2.2.1-15-2016-00007, GINOP-2.3.2-15-2016-00043 European Union and the Regional Development and Social Fund The work was supported by grants EFOP-3.6.1-16-2016-00022, GINOP-2.2.1-15-2016-00007, GINOP-2.3.2-15-2016-00043 from the National Research, Development and Innovation Office, and co-financed by the European Union and the Regional Development and Social Funds. The funders had no role in study design, data collection and analysis, decision to publish, or preparation of the manuscript.

==============================
The blood–brain barrier (BBB), an interface between the systemic circulation and the nervous system, can be a target of cytokines in inflammatory conditions. Pro-inflammatory cytokines tumor necrosis factor-α (TNF-α) and interleukin-1β (IL-1β) induce damage in brain endothelial cells and BBB dysfunction which contribute to neuronal injury. The neuroprotective effects of α-melanocyte stimulating hormone (α-MSH) were investigated in experimental models, but there are no data related to the BBB. Based on our recent study, in which α-MSH reduced barrier dysfunction in human intestinal epithelial cells induced by TNF-α and IL-1β, we hypothesized a protective effect of α-MSH on brain endothelial cells. We examined the effect of these two pro-inflammatory cytokines, and the neuropeptide α-MSH on a culture model of the BBB, primary rat brain endothelial cells co-cultured with rat brain pericytes and glial cells. We demonstrated the expression of melanocortin-1 receptor in isolated rat brain microvessels and cultured brain endothelial cells by RT-PCR and immunohistochemistry. TNF-α and IL-1β induced cell damage, measured by impedance and MTT assay, which was attenuated by α-MSH (1 and 10 pM). The peptide inhibited the cytokine-induced increase in brain endothelial permeability, and restored the morphological changes in cellular junctions visualized by immunostaining for claudin-5 and β-catenin. Elevated production of reactive oxygen species and the nuclear translocation of NF-κB were also reduced by α-MSH in brain endothelial cells stimulated by cytokines. We demonstrated for the first time the direct beneficial effect of α-MSH on cultured brain endothelial cells, indicating that this neurohormone may be protective at the BBB.

Introduction

The neuropeptide α-melanocyte stimulating hormone (α-MSH) belongs to the family of melanocortins, which are created from pro-opiomelanocortin (Catania et al., 2010). The α-MSH, a 13 amino acid long peptide hormone, is mainly produced in the hypothalamic region (Brzoska et al., 2008). Induction of melanogenesis in pigment cells and immunomodulation are among its main physiological functions (Catania, 2008). There are five melanocortin receptors (MCRs), from which four, MC1R, MC3R, MC4R, and MC5R, bind α-MSH (Brzoska et al., 2008). MCRs are G-protein-coupled and exert their effects via cyclic 3′,5′-adenosine monophosphate (cAMP)-dependent signaling pathways. The predominant receptor of α-MSH is MC1R, which binds the neuropeptide with high affinity, and is expressed in both the brain and periphery. MC1R was demonstrated in several tissues, such as brain, skin, immune system, and gut (Brzoska et al., 2008).

Melanocortins are evolutionary conserved defense molecules against tissue injury and bacterial invasion (Catania, 2008). The immunomodulatory and anti-inflammatory effects of α-MSH are well known and widely investigated. The anti-inflammatory action of α-MSH was proven in animal models of systemic or local inflammation, like sepsis, arthritis, uveitis, dermatitis, pancreatitis, and colitis (Brzoska et al., 2008). The protective effect of α-MSH was also shown in ischemic conditions in the heart (Vecsernyés et al., 2003, 2017), retina (Varga et al., 2013), kidney and intestine (Brzoska et al., 2008). In addition to animal models of gut inflammation (Rajora et al., 1997a), the effects of α-MSH were examined on cell cultures. Our groups have recently described that pro-inflammatory cytokines tumor necrosis factor-α (TNF-α) and interleukin-1β (IL-1β) disrupted the barrier integrity of Caco-2 human intestinal epithelial cells, which was attenuated by α-MSH via the inhibition of the NF-κB pathway (Váradi et al., 2017).

In addition to systemic inflammatory conditions, the beneficial effect of α-MSH was also investigated in models of acute and chronic injuries of the central nervous system (CNS; Catania, 2008). Treatment with α-MSH was neuroprotective in cerebral ischemia, traumatic spinal cord injury, kainic acid induced excitotoxic brain damage, and lipopolysaccharide (LPS)-induced cerebral inflammation in animal studies. One of the main mechanisms of action of α-MSH is the down-regulation of pro-inflammatory cytokines TNF-α and/or IL-1β in blood and brain tissue, as it was described in LPS-induced brain inflammation (Rajora et al., 1997b) and unilateral middle cerebral artery occlusion (Huang & Tatro, 2002) in mice, and in kainic acid-induced brain damage in rats (Forslin Aronsson et al., 2007). Studies on cultured astrocytes, neurons, and microglia revealed that, similarly to the periphery, inhibition of the NF-κB pathway and downstream blockade of pro-inflammatory cytokine release and nitric oxide overproduction contribute to the protective effects of α-MSH (Catania, 2008).

Dysfunction of the blood–brain barrier (BBB) has been described in many systemic and CNS inflammatory diseases (Erickson & Banks, 2018) and plays a central role in the pathomechanism of many neurological diseases (Zhao et al., 2015; Liebner et al., 2018). TNF-α and IL-1β are the two most studied major pro-inflammatory cytokines in neuroinflammation observed in CNS pathologies (Sochocka, Diniz & Leszek, 2017; Liebner et al., 2018), and induced by systemic inflammation (Hoogland et al., 2015). The elevated levels of these two cytokines, confirmed in chronic neurodegenerative diseases (Sochocka, Diniz & Leszek, 2017), open the BBB (Liebner et al., 2018). TNF-α directly increases the permeability of the BBB for marker molecules in animal studies (Megyeri et al., 1992; Ábrahám et al., 1996), as well as in culture models (Deli et al., 1995) and a similar effect was described for IL-1β in BBB in vitro experiments (for review see Deli et al., 2005). Since these two pro-inflammatory cytokines are major players in both systemic and CNS inflammation and induce BBB dysfunction, TNF-α and IL-1β were selected for our experiments.

A binding site was found on murine brain endothelial cells by radiolabeled α-MSH (de Angelis et al., 1995), suggesting the presence of MCR(s) at the BBB, but they were not identified. MCRs expressed on cells of the neurovascular unit could mediate the neuroprotective actions of melanocortins (Catania, 2008), however, the effects of α-MSH on the BBB, especially on brain endothelial cells have not been investigated yet.

Therefore, our aim was to investigate the effects of α-MSH peptide on a culture model of the BBB, its possible protective action against barrier disruption induced by TNF-α and IL-1β cytokines by measuring cellular viability, permeability for marker molecules, structure of interendothelial tight junctions, production of reactive oxygen species (ROS) and the nuclear translocation of NF-κB p65 subunit.

Materials and Methods

Materials

All reagents were purchased from Sigma-Aldrich Corporation (subsidiary of Merck KGaA, Darmstadt, Germany) unless otherwise indicated.

Cell culture

Primary rat brain endothelial cells were isolated from three week-old outbred Wistar rats (Harlan Laboratories, Indianapolis, IN, USA) as described in our previous paper (Veszelka et al., 2013). Forebrains were collected in ice-cold sterile phosphate buffered saline (PBS). After meninges were removed the tissue was cut into 1 mm3 pieces by scalpel and digested with enzymes (1 mg/ml collagenase type II, and 15 μg/ml deoxyribonuclease type I; Roche, Basel, Switzerland) in Dulbecco’s modified Eagle medium (DMEM/F12, Gibco; Life Technologies, Carlsbad, CA, USA) at 37 °C for 55 min. Microvessels were separated from myelin rich fraction by centrifugation in 20% BSA–DMEM gradient (1,000×g, 20 min, three times). The collected vessels were further digested with enzymes (1 mg/ml collagenase–dispase, and 15 μg/ml deoxyribonuclease type I; Roche, Basel, Switzerland) in DMEM/F12 at 37 °C for 35 min. Brain microvascular endothelial cell clusters were separated on a 33% continuous Percoll gradient (1,000×g, 10 min), collected, and washed twice in DMEM/F12. Cells were seeded onto Petri dishes (100 mm; Orange Scientific, Braine-l’Alleud, Belgium) coated with collagen type IV and fibronectin (100 μg/ml each). Cultures were maintained in DMEM/F12 supplemented with 15% plasma-derived bovine serum (First Link, Wolverhampton, UK), 5 μg/ml insulin-transferrin-sodium selenite (Pan Biotech, Aidenbach, Germany), 1 ng/ml basic fibroblast growth factor (Roche, Basel, Switzerland), 100 μg/ml heparin, and 5 μg/ml gentamycin. For the first three days of culture cells were grown in medium with 3 μg/ml puromycin to eliminate P-glycoprotein negative contaminating cell types (Perrière et al., 2005). When endothelial cells reached 90% confluency, they were subcultivated for different experiments.

Primary rat brain pericytes were isolated by the same method, except that the second digestion lasted only for 15 min. After isolation cells were plated onto uncoated dishes and they were cultured in low-glucose DMEM medium (Gibco, Life Technologies, Carlsbad, CA, USA) supplemented with 10% FBS (Pan Biotech, Aidenbach, Germany) and gentamycin. No puromycin was applied. Pericytes were used at third passage for experiments.

Primary rat glial cells were isolated from one-day-old Wistar rats. After meninges were removed brain tissue was mechanically dissociated by a syringe equipped with a long needle. The tissue homogenate was filtered through a nylon mesh (40 μm; Millipore, Billerica, MA, USA) to remove large tissue pieces and vessels. Cell clusters in the filtrate were plated onto uncoated 75 cm2 flasks (TPP, Trasadingen, Switzerland), cultured until 90% confluency in DMEM containing 10% FBS (Lonza, Basel, Switzerland) and gentamycin. For the BBB co-culture model glial cells were passaged at a cell number of 5 × 104 cells/well for 12-well plates (Corning Costar, Corning, NY, USA) and cultured for two weeks before use. Confluent glia cultures contained 90% of GFAP immunopositive astroglia, and 10 % CD11b immunopositive microglia.

To induce BBB characteristics brain endothelial cells were co-cultured with brain pericytes and glial cells (Nakagawa et al., 2009) using 12-well format tissue culture inserts (Transwell clear, polyester membrane, 0.4 μm pore size, 1.12 cm2 surface; Corning Costar, Corning, NY, USA). Rat pericytes (1.5 × 104 cells/cm2) were subcultivated to the bottom side of the insert membranes and brain endothelial cells (7.5 × 104 cells/cm2) were passaged to the upper side of the collagen type IV and fibronectin coated inserts. Both compartments received endothelial culture medium. After two days of co-culture, brain endothelial cells reached confluency and 550 nM hydrocortisone was added to the culture medium to tighten junctions (Walter et al., 2015). Before experiments 250 μM 8-(4-chlorophenylthio)-cAMP and 17.5 μM Ro 20-1724 (Roche, Basel, Switzerland) were added to the endothelial cells for 24 h to tighten junctions and elevate resistance (Deli et al., 2005; Perrière et al., 2005).

Treatments

The neuropeptide α-MSH (Mw. 1664.8 Da) was tested at 1 pM to 1 μM concentrations. The use of cytokine cocktails is common in CNS culture studies (Sochocka, Diniz & Leszek, 2017). We have also used a combination of TNF-α (50 ng/ml) and IL-1β (25 ng/ml) on intestinal epithelial cells in our previous study (Váradi et al., 2017). Based on this study, we have tested a combination of the pro-inflammatory cytokines TNF-α (10–50 ng/ml) and IL-1β (10–25 ng/ml) for 24 h to induce damage in brain endothelial cells. The control group received culture medium only.

Total RNA isolation and RT-PCR

Brain microvessels were isolated from adult rat brains as published earlier (Veszelka et al., 2007). For receptor expression analysis total RNA was isolated from isolated brain microvessels and brain endothelial cells by TRI Reagent (Molecular Research Center, Cincinnati, OH, USA). One microgram of total RNA was transcribed to cDNA by Maxima First Strand cDNA Synthesis Kit (ThermoFisher, Waltham, MA, USA). Specific oligonucleotide primers were designed for the Mc1r gene (XM_006222790). Primer sets were MC1R_fwd 5′-TGCACCTCTTGCTCATCGTT-3′ and MC1R_rvs 5′-ACCTCCTTGAGTGTCATGCG-3′. The predicted length of PCR product was 160 bps. Primers for β-actin were used as internal controls (NM_031144). Primer sets were ACT_fwd 5′-TACTCTGTGTGGATTGGTGGC-3′ and ACT_rvs 5′-GGTGTAAAACGCAG CTCAGTAA-3′. The predicted length of PCR product was 150 bps. PCR was performed with FIREPol DNA polymerase (Solis BioDyne, Tartu, Estonia) in T100 thermal cycler (BioRad, Hercules, CA, USA). After initial heat inactivation (95 °C for 3 min) the following cycling conditions were applied: melting 94 °C for 15 s, annealing 50 °C for 15 s, polimerization 72 °C for 20 s (35 cycles). After a final 5 min extension at 72 °C PCR products were analyzed on 3% MetaPhor agarose gel (Cambrex BioScience, Rockland, ME, USA) and fragments were verified by capillary DNA sequencing.

Measurement of cell viability: real-time cell electronic sensing and MTT assay

Real-time cell electronic sensing is an impedance-based, label-free technique for dynamic monitoring of living adherent cells, including barrier forming epithelial and brain endothelial cells (Kiss et al., 2014; Lénárt et al., 2015). The RTCA-SP instrument (ACEA Biosciences, San Diego, CA, USA) registers the impedance of cells in every 10 min and for each time point cell index is calculated as (Rn − Rb)/15, where Rn is the cell-electrode impedance of the well when it contains cells and Rb is the background impedance of the well with the medium alone. E-plates, special 96-well plates with built in gold electrodes, were coated with collagen type IV and fibronectin for brain endothelial cells at room temperature and dried for 20 min under UV and air-flow. For background measurements culture medium (60 μl) was added to each well, then 50 μl of rat brain endothelial cell suspension was distributed at a cell density of 5 × 103 cells/well. After cells reached a steady growth phase they were treated with α-MSH and cytokines.

In parallel we also used an endpoint colorimetric cell viability assay (Kiss et al., 2014). The yellow 3-(4,5-dimethylthiazol-2-yl)-2,5-diphenyltetrazolium bromide (MTT) dye is converted by viable cells to purple formazan crystals. Brain endothelial cells were grown in 96-well plates coated with collagen type IV and fibronectin. After treatment of confluent monolayers, cells were incubated with 0.5 mg/ml MTT solution in cell culture medium for 3 h in CO2 incubator. Formazan crystals were dissolved in dimethyl sulfoxide, and dye concentration was determined by absorbance measurement at 570 nm with a multiwell plate reader (Fluostar Optima; BMG Labtechnologies, Offenburg, Germany).

Measurement of permeability for marker molecules

Penetration of fluorescein isothiocyanate-labeled dextran (FITC-dextran, MW: 4.4 kDa) and Evans blue-labeled albumin (EBA, MW: 67 kDa) across the BBB model was determined in permeability studies (Veszelka et al., 2007). Rat brain endothelial cells co-cultured with pericytes and glia on cell culture inserts were treated with cytokines and/or α-MSH for 1 h. After treatment inserts were transferred to 12-well plates containing 1.5 ml Ringer-HEPES buffer (150 mM NaCl, 2.2 mM CaCl2, 0.2 mM MgCl2, 5.2 mM KCl, 6 mM NaHCO3, 5 mM glucose, and 10 mM HEPES, pH 7.4) in the lower compartments. In the upper compartments culture medium was replaced by 500 μl Ringer-HEPES buffer containing EBA complex (1 mg/ml bovine serum albumin and 167.5 μg/ml Evans blue) and FITC-dextran (100 μg/ml). The plates were kept for 30 min on a rocking platform (100 rpm) at 37 °C in an incubator with 5% CO2. After the incubation the concentrations of the marker molecules in samples from the upper and lower compartments were determined by a fluorescence multiwell plate reader (Fluostar Optima; BMG Labtechnologies, Offenburg, Germany; excitation wavelength: 485 nm, emission wavelength: 520 nm for FITC-dextran; excitation wavelength: 584 nm, emission wavelength: 680 nm for EBA). The apparent permeability coefficients (Papp) were calculated by the following equation (Váradi et al., 2017): (1) Papp =[C]B ×  VBA×[C]A×t

where [C]B is the concentration of the tracer in the lower (basal) compartment after 1 h, [C]A is the concentration of the tracer in the upper (apical) compartment at 0 h, VB is the volume of the basal compartment (1.5 ml) and A is the surface area available for permeability (1.12 cm2).

Immunohistochemistry

Melanocortin-1 receptor

Isolated rat brain microvessels and brain endothelial cells, co-cultured with brain pericytes and glial cells, were immunostained for MC1R as we described in our previous paper for epithelial cells (Váradi et al., 2017). Cells were washed in PBS, fixed with ice cold acetone/methanol solution (1:1) for 5 min, and blocked with 3% BSA–PBS for 1 h. A rabbit anti-human MC1R antibody (M9193; 1 mg/ml) was used as primary antibody for overnight at 4 °C. Incubation with secondary antibody, Alexa Fluor 488 conjugated goat-anti-rabbit IgG (A11029; ThermoFisher Scientific, Waltham, MA, USA; 2 mg/ml) and nucleus stain ethidium homodimer-1 (1 μM) lasted for 1 h. The samples were thoroughly washed with PBS after each step and mounted in Fluoromount-G (Southern Biotech, Birmingham, AL, USA). Stainings were visualized by a Leica SP5 confocal laser scanning microscope (Leica Microsystems GmbH, Wetzlar, Germany).

Junctional proteins and NF-κB p65 subunit

Rat brain endothelial cells treated with cytokines and/or α-MSH for 1 h were fixed and permeabilized with ice cold acetone/methanol solution (1:1) and blocked in 3% BSA–PBS for 1 h. Cell were incubated with rabbit anti-rat β-catenin polyclonal (C2206; 1 mg/ml) or rabbit anti-rat claudin-5 polyclonal (SAB4502981; 1 mg/ml) primary antibodies overnight, followed by secondary labeling with Cy3-conjugated goat-anti-rabbit IgG (C2306; 2 mg/ml) and a nucleus stain (H33342; 1 μg/ml) for 1 h. For NF-κB p65 subunit immunostaining a rabbit anti-human p65 polyclonal antibody (sc-372; Santa Cruz Biotechnology, Dallas, TX, USA; 100 μg/ml, overnight incubation) was used, followed by incubation with Alexa Fluor 488-conjugated goat-anti-rabbit IgG (A11029; ThermoFisher Scientific, Waltham, MA, USA; 2 mg/ml) and H33342 for 1 h. Cells were washed and mounted as described above. For both TJ protein and NF-κB stainings samples were observed by a Leica SP5 confocal laser scanning microscope. Images showing TJ staining were analyzed using MATLAB software (MathWorks, Natick, MA, USA). The backgrounds of each image were determined and subtracted from the corresponding image to compensate the occasional non-uniform background. Then, the grayscale images were converted to binary. Objects with size less than 6 pixels were eliminated to reduce any false structure. The areas of the structures were determined by the pixel number of the structures on the binary images. The object number defines the number of the separated, non-contact structure elements of the images. Images of NF-κB immunostaining were analyzed using ImageJ software (National Institutes of Health, Rockville, MD, USA) as described in our previous paper (Sántha et al., 2016).

Measurement of ROS production

Reactive oxygen species production was detected by chloromethyl-dichloro-dihydro-fluorecein diacetate (DCFDA; Life Technologies, Carlsbad, CA, USA) as we described previously (Veszelka et al., 2013; Lénárt et al., 2015). The indicator penetrates into the cells and interacts with intracellular esterases. After its oxidation fluorescent molecules are produced, and the fluorescent signal is proportional to the produced ROS. Confluent brain endothelial cells were cultured in black 96-well plates with glass bottoms (Corning Costar, Corning, NY, USA). After 1 h treatment cells were incubated with Ringer-HEPES buffer containing 2 μM DCFDA for 1 h at 37 °C. Hydrogen peroxide (100 μM) was used as a reference compound (positive control). Fluorescence was measured at 485 nm excitation and 520 nm emission wavelengths by a Fluostar Optima multiwell plate reader (BMG Labtechnologies, Offenburg, Germany) every 5 min for 1 h. Fluorescence intensities are shown in arbitrary units.

Statistical analysis

Statistical analysis was done by GraphPad Prism 5.0 software (GraphPad Software Inc., La Jolla, CA, USA). Data are presented as mean ± SEM. Comparison of groups was performed using ANOVA and Dunnett or Bonferroni tests. Differences were considered significant at P < 0.05.

Results

Expression of melanocortin-1 receptor on rat brain microvessels and cultured brain endothelial cells

MC1R gene expression was detected on both freshly isolated rat brain microvessels and confluent rat brain endothelial cell cultures with reverse transcription followed by PCR amplification using the specifically designed rat primers (Fig. 1A). This result was strengthened by immunohistochemistry for MC1R. The brain endothelial staining (Fig. 1B) was similar to the apical surface staining of MC1R on Caco-2 cells in our previous work (Váradi et al., 2017). Positive MC1R immunostaining was found on freshly isolated brain microvessels, too (Fig. 1C).

Figure 1 Melanocortin-1 receptor (MC1R) expression on rat brain endothelial cells and isolated rat brain microvessels.

(A) Cells were grown in culture dish and brain microvessels were isolated from adult rat brain. Total RNA was extracted, and reverse transcription and PCR amplification were performed for Mc1r. Actin gene was used as reference. The predicted length of the PCR products was 160 bps. Fragments were analyzed on MetaPhor agarose gel (3%) next low range DNA marker. (B) Cultured brain endothelial cells were grown on culture inserts and immunostained for MC1R. Cell nuclei were labeled with ethidium homodimer-1 (red). Green: MC1R labeling. Scale bar: 10 μm. (C) Isolated brain microvessels were immunostained for MC1R. Fluorescent labeling is shown together with differential interference contrast to reveal the structure of brain capillary. Cell nuclei were labeled with ethidium homodimer-1 (red). Green: MC1R labeling. Scale bar: 5 μm. MC1R, melanocortin-1 receptor; RBEC, rat brain endothelial cells; MV, microvessels.

Effect of α-MSH on rat brain endothelial cell metabolism and viability

Treatment of rat brain endothelial cells with α-MSH peptide alone (1 pM to 1 μM) did not have any significant effect on cell kinetics measured by impedance, indicating good cell viability, attachment and barrier integrity. Followed by real-time cell electronic sensing the cells showed no change after α-MSH treatment, the kinetic curves ran similarly to the untreated control samples (Fig. 2A). In addition, the colorimetric MTT assay also did not detect any metabolic change in cells after 24 h treatment with α-MSH as compared to control (Fig. 2B). For further experiments 1 and 10 pM concentrations of α-MSH were selected, which correspond to the physiological range of the neurohormone in the blood (Kovács et al., 2001; Magnoni et al., 2003).

Figure 2 The effect of different concentrations of α-MSH on the cell viability of rat brain endothelial cells.

(A) Cultured brain endothelial cells were treated with α-MSH (1 pM to 1 μM) for 24 h. Control group received culture medium. Endothelial cells treated with 1% Triton X-100 detergent were used as cytotoxicity control. The cell index followed by impedance did not change after α-MSH treatment. (B) The metabolic activity of rat brain endothelial cells was measured by MTT assay, which did not detect any alteration caused by α-MSH. Mean ± SEM, n = 4–8. C, control group; TX-100, Triton X-100.

Effect of α-MSH on cytokine treated rat brain endothelial barrier integrity: real-time cell kinetics

To determine the treatment concentrations of TNF-α and IL-1β to induce brain endothelial damage, four different combinations of cytokine concentrations (10 + 10, 10 + 25, 25 + 25, 50 + 25 ng/ml) were tested (Fig. 3A). The cytokine combination in the tested concentration range decreased brain endothelial viability. There were no statistically significant differences between the four cytokine treatment groups. For further experiments the smallest, 10 + 10 ng/ml concentrations of the cytokines were selected. Treatment of rat brain endothelial cells with TNF-α and IL-1β decreased the cell index by more than 50% at 6 h time point which further dropped by 24 h (Fig. 3B). Low concentration of α-MSH (1 pM) significantly protected against the cytokine-induced cell viability decrease (Fig. 3C). There was no statistically significant difference between the two cytokine + α-MSH treatment groups.

Figure 3 The effect of α-MSH treatment on cell viability of cytokine treated rat brain endothelial cells.

(A) Rat brain endothelial cells were treated with four different combinations of cytokine concentrations (TNF-α + IL-1β: 10 + 10, 10 + 25, 25 + 25, and 50 + 25 ng/ml) for 24 h and the cellular effects were monitored by impedance. Control group received culture medium. (B) Rat brain endothelial cells were treated with cytokines (10 ng/ml TNF-α and 10 ng/ml IL-1β) without or with α-MSH (1 and 10 pM) and the cellular effects were monitored by impedance for 24 h. Control group received culture medium. The cytokines decreased the cell index, which effect could be ameliorated by α-MSH treatment, especially by the lower, 1 pM α-MSH concentration. (C) After 24 h treatment the cell viability significantly decreased due to cytokine treatment, while 1 pM α-MSH significantly blocked the cytokine effect. Mean ± SEM, n = 3–6, ***P < 0.001, #P < 0.05. Asterisks indicate that groups were compared to the control group. Pound signs indicate that groups were compared to the cytokine-treated group. C, control group; CK, cytokine treated group.

Effect of α-MSH on cytokine treated rat brain endothelial barrier integrity: permeability for marker molecules

The integrity of brain endothelial monolayers was tested by its permeability for marker molecules. A low permeability for both FITC-dextran (1.91 ± 0.02 × 10−6 cm/s) and large biomolecule albumin (0.88 ± 0.13 × 10−6 cm/s) was measured on the triple co-culture model of the BBB. Cytokine treatment significantly increased the permeability of the co-cultures for both markers (Fig. 4). The lower, 1 pM concentration of α-MSH peptide significantly blocked the barrier opening effect of the cytokines for FITC-dextran and albumin, while 10 pM α-MSH significantly strengthened the cellular barrier for the bigger marker molecule.

Figure 4 The effect of α-MSH treatment on the permeability of cytokine treated triple co-culture model of the BBB.

Brain endothelial cells co-cultured with pericytes and astrocytes on culture inserts were treated with cytokines (10 ng/ml TNF-α and 10 ng/ml IL-1β) without or with α-MSH (1 or 10 pM) for 1 h. Then, permeability was measured for 4 kDa FITC-dextran and albumin. Control group received culture medium. Mean ± SEM, n = 7–17, ***P < 0.001. Asterisks indicate that groups were compared to the control group. Pound signs indicate that groups were compared to the cytokine-treated group. C, control group; CK, cytokine treated group; ns, no significant difference between the two groups.

Effect of α-MSH on cytokine treated rat brain endothelial barrier integrity: immunostaining for claudin-5 and β-catenin junctional proteins

Rat brain endothelial cells form tight paracellular barrier, which was visualized by the localization of integral membrane TJ protein claudin-5 and adherens junction protein β-catenin (Fig. 5). These junctional proteins appeared in the cell membranes in a continuous, belt-like manner. In the control, untreated group continuous stainings were visible without gaps at the cell border. In the cytokine treated group gaps and fragmented junctional staining (arrowheads, Fig. 5A) and cytoplasmic redistribution of junctional proteins (asterisks, Fig. 5A) were observed. In cells treated with both α-MSH and cytokine the staining of TJ proteins was more similar to the control group. The object number on the immunostained pictures significantly increased due to cytokine treatment as compared to the control and α-MSH treated group (Figs. 5B and 5C).

Figure 5 The effect of α-MSH treatment on the immunostaining of tight junction proteins claudin-5 and β-catenin in cytokine treated rat brain endothelial cells.

(A) Rat brain endothelial cells were treated with cytokines (10 ng/ml TNF-α and 10 ng/ml IL-1β) without or with 1 pM α-MSH for 1 h and stained for tight junction proteins claudin-5 and β-catenin. The control group received culture medium. The fluorescent microscopy images demonstrate the expression and organization of the junctional proteins. Cytokine treatment significantly increased the number of gaps and intracellular redistribution compared to the control and α-MSH treated group. Arrowheads: gaps and fragmented junctional staining; asterisks: cytoplasmic redistribution of junctional proteins. Scale bar: 10 μm. (B) The object number on the claudin-5 immunostained pictures was quantified by MATLAB software. (C) The object number on the β-catenin immunostained pictures was quantified by MATLAB software. Mean ± SEM, n = 4, *P < 0.05, #P < 0.05. *: CK compared to C; #: CK + MSH compared to CK. C, control group; CK, cytokine treated group; α-MSH + CK, α-MSH, and cytokine treated group.

Effect of α-MSH on cytokine treated rat brain endothelial cells: ROS production

Treatment with TNF-α and IL-1β significantly increased ROS production in brain endothelial cells compared to the basal ROS production of the control group (Fig. 6). Low concentrations of α-MSH peptide alone did not change the basal ROS production, while it could statistically significantly decrease the cytokine-induced ROS production in cultured brain endothelial cells. Hydrogen peroxide treatment was used as a reference inducer of ROS in the assay, as in our previous study (Veszelka et al., 2013).

Figure 6 The effect of α-MSH treatment on the reactive oxygen species production in cytokine treated rat brain endothelial cells.

Rat brain endothelial cells were treated with cytokines (10 ng/ml TNF-α and 10 ng/ml IL-1β) without or with 1 or 10 pM α-MSH for 1 h. Cells were also treated with 1 or 10 pM α-MSH alone. Control group received culture medium. Mean ± SEM, n = 4–8. C, control group; CK, cytokine treated group; H2O2, hydrogen peroxide treated group (100 μM). **P < 0.01, ***P < 0.001, #P < 0.05, ###P < 0.001. Asterisks indicate that groups were compared to the control group. Pound signs indicate that groups were compared to the cytokine-treated group. C, control group; CK, cytokine treated group; α-MSH+CK, α-MSH and cytokine treated group.

Effect of α-MSH on cytokine-induced NF-κB nuclear translocation in rat brain endothelial cells

Cytokine treatment induced inflammatory reaction can be measured reliably by the translocation of the p65 subunit of NF-κB transcriptional factor into cell nuclei. In the control group the localization of NF-κB p65 is mostly cytoplasmic (Fig. 7A). Treatment with inflammatory cytokines induced the nuclear translocation of the transcription factor in rat brain endothelial cells (Fig. 7). This NF-κB nuclear translocation, which was also quantified based on the fluorescent intensity of the immunostainings, was inhibited by α-MSH at 1 pM concentration (Fig. 7A), and which was the most effective in other experiments.

Figure 7 Immunohistochemical staining and analysis of NF-κB activation in cytokine- and α-MSH-treated brain endothelial cells.

(A) Rat brain endothelial cells were treated with culture medium, cytokines (10 ng/ml TNF-α and 10 ng/ml IL-1β), or cytokines and 1 pM α-MSH. Nuclear localization of the NF-κB p65 subunit was monitored by immunostaining. Cell nuclei were labeled with Hoechst 33342. Green: p65 staining; blue: cell nuclei. Scale bar: 25 μm. (B) Fluorescence intensity of the NF-κB immunostaining in cell nuclei and cytoplasm. Mean ± SEM, n = 20–28, *P < 0.05, #P < 0.05. *: CK, CK + MSH compared to C; #: CK + MSH compared to CK. C, control group; CK, cytokine treated group; α-MSH + CK, α-MSH and cytokine treated group.

Discussion

The proper function of the BBB is crucial for CNS homeostasis and BBB dysfunction can be both the cause and consequence of neuronal injury (Stanimirovic & Friedman, 2012; Zhao et al., 2015; Liebner et al., 2018). The neuroprotective effects of α-MSH were already investigated in different in vivo experimental models (Catania, 2008), but there are no data related to the BBB. Based on our recent study, in which α-MSH reduced cytokine-induced barrier dysfunction in human intestinal epithelial cells (Váradi et al., 2017), we hypothesized a protective effect of this neurohormone on brain endothelial cells.

MC1R participates in the mediation of the anti-inflammatory action of α-MSH in both the CNS and the periphery (Brzoska et al., 2008). We demonstrated for the first time the expression of MC1R, the main receptor of α-MSH, on isolated rat brain microvessels and cultured brain endothelial cells both at the mRNA and protein level (Fig. 1). MC1R is expressed not only in the CNS but also in peripheral tissues (Brzoska et al., 2008), including human intestinal epithelial cells (Váradi et al., 2017). Our findings are in agreement with the presence of α-MSH binding sites on cultured mouse brain endothelial cells (de Angelis et al., 1995).

We tested the effect of α-MSH on rat brain endothelial cells in a wide range of concentrations, from 1 pM to 1 μM, and found no change in the cell kinetics indicating undisturbed cell function and monolayer integrity (Fig. 2). The physiological concentration of α-MSH is low in both human plasma (6 pM; Magnoni et al., 2003) and newborn pig plasma samples (30 pM; Kovács et al., 2001). The level of α-MSH changes in pathological conditions: in our previous study asphyxia and reperfusion increased the level of α-MSH in the plasma, but decreased it in the cerebrospinal fluid of newborn pigs (Kovács et al., 2001). The neuropeptide in the blood is secreted from the pituitary, while α-MSH in the cerebrospinal fluid is derived from the CNS, explaining the discordant regulation of the neuropeptide in these two compartments (de Rotte, Bouman & van Wimersma Greidanus, 1980). A low permeability for intravenously injected α-MSH was measured across the BBB in rats (Wilson, 1988), indicating the absence of an active transport. At pathologically high or pharmacological concentrations α-MSH crosses the BBB contributing to the brain levels (de Rotte, Bouman & van Wimersma Greidanus, 1980; Banks & Kastin, 1995).

To mimic inflammatory conditions, a combination of TNF-α and IL-1β was selected to induce barrier dysfunction on the BBB model, similarly to our previous study on cultured intestinal epithelial cells (Váradi et al., 2017). Brain endothelial cells were more sensitive to cytokine treatment, lower concentrations were already effective to induce enhanced permeability as compared to epithelial cells (Váradi et al., 2017). Pro-inflammatory cytokines are linked to increased permeability in brain endothelial cell cultures (Deli et al., 1995; Didier et al., 2003; Lopez-Ramirez et al., 2012). As expected, TNF-α and IL-1β decreased the impedance of the cell layers (Fig. 3) and elevated the permeability for marker molecules dextran and albumin on the BBB model (Fig. 4), indicating a barrier leakage, which was inhibited by α-MSH. The same barrier protective effect was found on intestinal epithelial cells, too (Váradi et al., 2017). Treatment with α-MSH in physiological concentrations did not modify the basal permeability of the BBB model. Previous studies in rats also found that α-MSH at low concentrations did not alter BBB permeability for albumin (Sankar, Domer & Kastin, 1981) or a small anion (Kastin & Fabre, 1982).

The cytokine-induced disruption of rat brain endothelial cell layers was also reflected at the level of cell–cell interactions. In concordance with the functional measurements of barrier opening, cytokine treatment caused a discontinuous immunostaining at cellular borders, and cytoplasmic redistribution for claudin-5 and β-catenin (Fig. 5). The fragmented staining at the intercellular junctions and the increased presence of the junctional proteins in the cytoplasm of brain endothelial cells were reflected by the elevated object number on the immunostained culture samples. In our recent work we also demonstrated morphological changes at cellular junctions in parallel with opening of the barrier by TJ modulator peptides in a BBB model (Bocsik et al., 2016). Elevated TNF-α and IL-1β expression, disruption of TJ proteins and increased BBB permeability were also found in mice treated with an environmental toxicant and nanoparticles (Zhang et al., 2012). The direct link between these events was proven in two independent works. NF-κB response elements were identified in the promoter region of claudin-5 gene, and TNF-α treatment significantly reduced the promoter activity and the transcription of claudin-5 in both brain and myocardial endothelial cells (Burek & Förster, 2009). TNF-α treatment repressed claudin-5 promoter activity in mouse brain endothelial cell cultures via NF-κB signaling and p65 overexpression (Aslam et al., 2012). In addition to claudin-5, the decrease of MARVELD-2, another TJ protein was also induced by pro-inflammatory cytokine treatment in human brain endothelial cells (Lopez-Ramirez et al., 2013).

Oxidative stress is one of the main mechanisms of neuronal toxicity in inflammation (Sochocka, Diniz & Leszek, 2017). ROS are also central in increased BBB permeability induced by cytokines (Rochfort & Cummins, 2015), or by amyloid-β peptide (Veszelka et al., 2013). We also measured elevated ROS production after cytokine treatment in brain endothelial cells, which was inhibited by α-MSH (Fig. 6). While no data are available on BBB models, suppression of ROS production and oxidative stress as an important element of the anti-inflammatory effect of α-MSH were demonstrated on other cell types (Brzoska et al., 2008).

Translocation of the NF-κB p65 subunit into cell nuclei is a key event in inflammatory reactions in the CNS (Catania, 2008). The canonical NF-κB pathway can be induced by both TNF-α and IL-1β and lead to the transcription of genes, like cyclooxigenase-2, nitric oxide synthase, inflammatory cytokines, and matrix metalloproteinases (Pires et al., 2018), which participate in BBB dysfunction and opening (Rosenberg, 2012). The nuclear translocation of NF-κB was described in peripheral endothelial cells induced by TNF-α or IL-1β (Makó et al., 2010), as well as in brain endothelial cell inflammatory pathway (Lee et al., 2001). We confirmed in our study that cytokine treatment induced NF-κB nuclear translocation in brain endothelial cells, and proved that α-MSH can inhibit this effect (Fig. 7), similarly to our recent findings on epithelial cells (Váradi et al., 2017). Since NF-κB signaling in TNF-α treated brain endothelial cells can directly decrease claudin-5 expression (Burek & Förster, 2009; Aslam et al., 2012), this signaling pathway can mediate, at least partially, the protective effect of α-MSH on barrier integrity. Our data are supported by another study where α-MSH blocked adhesion molecule expression by NF-κB inhibition in cytokine treated human dermal microvascular endothelial cells (Kalden et al., 1999). These results further strengthen the role of α-MSH as a potent anti-inflammatory molecule which exerts its effect by inhibition of the NF-κB transcription factor.

In conclusion, the direct protective effect of α-MSH on pro-inflammatory cytokine-induced barrier dysfunction and inflammatory activation in rat brain endothelial cell cultures was investigated for the first time in this study. We demonstrated the presence of the major α-MSH receptor MC1R on brain endothelial cells, and a protective effect of the anti-inflammatory hormone on cytokine-induced barrier opening in parallel with the inhibition of NF-κB nuclear translocation in a BBB model. These findings support the beneficial effect of α-MSH to restore BBB integrity in inflammatory conditions.

Supplemental Information

Supplemental Information 1 Raw data obtained in the experiments.

Click here for additional data file.

Supplemental Information 2 Full length uncropped gel for Fig. 1.

PCR products for MC1R and beta-actin on a 3% MetaPhore gel.

Click here for additional data file.

Additional Information and Declarations

Competing Interests

Author Contributions

Data Availability

Maria A. Deli is an Academic Editor for PeerJ. Vilmos Tubak is employed by Creative Laboratory Ltd.

András Harazin conceived and designed the experiments, performed the experiments, analyzed the data, prepared figures and/or tables, authored or reviewed drafts of the paper, approved the final draft.

Alexandra Bocsik conceived and designed the experiments, performed the experiments, analyzed the data, prepared figures and/or tables, authored or reviewed drafts of the paper, approved the final draft.

Lilla Barna performed the experiments, analyzed the data, prepared figures and/or tables, approved the final draft.

András Kincses performed the experiments, analyzed the data, prepared figures and/or tables, approved the final draft.

Judit Váradi conceived and designed the experiments, contributed reagents/materials/analysis tools, authored or reviewed drafts of the paper, approved the final draft.

Ferenc Fenyvesi conceived and designed the experiments, contributed reagents/materials/analysis tools, authored or reviewed drafts of the paper, approved the final draft.

Vilmos Tubak analyzed the data, contributed reagents/materials/analysis tools, authored or reviewed drafts of the paper, approved the final draft.

Maria A. Deli conceived and designed the experiments, analyzed the data, contributed reagents/materials/analysis tools, authored or reviewed drafts of the paper, approved the final draft.

Miklós Vecsernyés conceived and designed the experiments, contributed reagents/materials/analysis tools, authored or reviewed drafts of the paper, approved the final draft.

The following information was supplied regarding data availability:

The raw data are provided as a Supplemental File.

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
