# Peer review of "Protection of cultured brain endothelial cells from cytokine-induced damage by α-melanocyte stimulating hormone"

_PeerJ, doi:10.7717/peerj.4774_

## Round 0.1 · original submission · Minor Revisions

Dear Maria,

Three reviewers have provided their opinions although only one of them provided any suggestions for improvement. Please respond to all the comments of Reviewer 2.

·

Basic reporting

excellent study

Experimental design

excellent design

Validity of the findings

novel and inovative

Additional comments

Harazin and coauthors have investigated the protective effect of alpha-MSH on brain endothelial cells and in vitro BBB model system. They found that alpha-MSH attenuated the cytokines –induced- BBB permeability, reduced production of reactive oxygen species, and NF-kB translocation.

This is one excellent study. In general, I found the manuscript well prepared and study is carefully designed and mapped out. Data are convincing, well presented and discussed. I did not have any specific concerns regarding this study.

Reviewer 2 ·

Basic reporting

In this study ‘Protection of rat brain endothelial cells from cytokine-induced damage by alpha-melanocyte stimulating hormone’ by Harazin et al., the authors established an in vitro culture model of BBB and studied the protective effects of alpha-MSH on the integrity of BBB from cytokines IL-1beta and TNF-alpha induced damage. The study’s background, experimental flow and data analysis were clearly reported, and appropriate publications have been cited to support the author’s work.

Experimental design

The idea of using alpha-MSH to protect BBB from inflammation-induced damage is in general well tested in vitro. However, there are some limitations regarding the experimental set up and design. First, the authors used an overly simplified model of just two cytokines at one dose to mimic neuroinflammation. It is unclear why a fixed dose of cytokines were chosen without titration, and why the authors only included two cytokines while in reality many cytokines are involved in neuronflammation (i.e IL-6, IL-18, IL-33). The authors should confirm their studies with a more comprehensive cytokine cocktail and also try a series of doses to demonstrate the true efficacy of alpha-MSH.

The in vitro protective effect of alpha-MSH was modest in most experiments, therefore one would question its efficacy in vivo, when delivery route and other barriers may further limit the final amount of alpha-MSH that could actually exert effect on brain endothelial cells. The authors should set up at least some in vivo assays to show the protective effects of alpha-MSH in vivo. In addition, as the authors mentioned alpha-MSH has receptors on different tissues, the authors should carefully address the potential issue of toxicity/off-target effect of using alpha-MSH in vivo.

Validity of the findings

As mentioned above, the neuronal protective effect of alpha-MSH was modest in most experiments. Especially in the first in some assays, the effects at higher 10uM concentration were actually more modest than those at a lower dose (1uM) (i.e. data in Figure 3). The authors should at least discuss why alpha-MSH effect did not correspond well with the applied dose.

Figure 1: the authors only showed the expression of MC1R on cells cultured in vitro, which is somewhat artificially and could be modulated by culture conditions. It is better to actually do IHC or IF on tissue sections to demonstrate MC1R expression in vivo.

Figure 5: The difference of barrier integrity as shown by claudin-5 and beta-catenin expression as shown was at best modest. It is hard to tell the difference by looking at the picture and find where the junction was breached. It would be much better to quantify the data and point out where the damages are in the picture. Also the authors need to clarify how many pictures they actually quantify, more than one experiments should be done.

Reviewer 3 ·

Basic reporting

The article 'Protection of rat brain endothelial cells from 1 cytokine-induced damage by α-melanocyte
stimulating hormone' by Harazin et al, have explored the inhibitory role of MSH in neuroinflammation specifically in the context of blood brain barrier. In my opinion, the authors report important and translationally relevant implications of this biology that will be of interest to the field in general. The observations appear to be robust and have been effectively described. Overall, the link between MSH and blood brain barrier protection has been established through well-designed experiments in multiple settings, suggesting a general role for MSH in regulating blood brain barrier function. I have no major concerns. I strongly suggest the manuscript for publication.

Experimental design

No comment

Validity of the findings

No comment

---

## Round 0.2 · accepted · Accept

The manuscript is already for publication

#